# Impact of Washing with Antioxidant-Infused Soda–Saline Solution on Gel Functionality of Mackerel (*Auxis thazard*) Surimi

**DOI:** 10.3390/foods12173178

**Published:** 2023-08-24

**Authors:** Porntip Thongkam, Manat Chaijan, Ling-Zhi Cheong, Worawan Panpipat

**Affiliations:** 1Food Technology and Innovation Research Center of Excellence, School of Agricultural Technology and Food Industry, Walailak University, Nakhon Si Thammarat 80160, Thailand; pohntip23@gmail.com (P.T.); pworawan@wu.ac.th (W.P.); 2School of Agriculture and Food, Faculty of Veterinary and Agricultural Sciences, The University of Melbourne, Parkville, Melbourne, VIC 3010, Australia; lingzhi.cheong@unimelb.edu.au

**Keywords:** surimi, mackerel, antioxidant, gel, oxidation

## Abstract

Mackerel (*Auxis thazard*), a tropical dark-fleshed fish, has the potential to be used in the production of surimi. It is necessary to identify the optimal washing method to make better use of this species since efficient washing is the most important step in surimi processing to ensure maximal gelling and high-quality surimi. The purpose of this study was to evaluate the combined effect of cold carbonated water (CW) with NaCl and antioxidants in washing media, so-called antioxidant-infused soda–saline solution, on lipid and myoglobin removal efficacy, biochemical characteristics, gelling properties, sensory features, and the oxidative stability of mackerel surimi in comparison with unwashed mince (T1) and conventional water washed surimi (T2). Mackerel mince was washed with CW in the presence of 0.6% NaCl at a medium to mince ratio of 3:1 (*v*/*w*) without antioxidant (T3) or with the addition of 1.5 mM EDTA plus 0.2% (*w*/*v*) sodium erythorbate and 0.2% sodium tripolyphosphate (T4), 100 mg/L gallic acid (T5), and 5 mM citric acid containing 8 mM calcium chloride (T6). During the first washing cycle, the antioxidants were mixed into the washing medium. The second and third washing cycles were then completed with cold water. The yields of all treatments were roughly 75–83%, based on the gross weight of the raw mince. The pH of the surimi was in a range of 5.47–6.46. All of the surimi had higher reactive sulfhydryl (SH) content and surface hydrophobicity but lower Ca^2+^-ATPase activity than unwashed mince (*p* < 0.05). After washing, lipids decreased significantly (*p* < 0.05), accounted for a 65–76% reduction. The T2 surimi had the highest peroxide value (PV). T1 had the lowest conjugated diene value. T1 and T4 surimi had the lowest TBARS value (*p* < 0.05). A lower non-heme iron level was found in all antioxidant-treated samples than in T1. Washing can increase the redox stability of myoglobin regardless of the washing media, as seen by the relatively low metmyoglobin levels. According to the dynamic viscoelastic behavior, all surimi and unwashed mince underwent the same degree of sol–gel transition following heat gelation. T1 showed the lowest breaking force, deformation, gel strength, and whiteness (*p* < 0.05). Surimi made from T4 or T5 had the highest gel strength when both breaking and deformation were considered, but the latter’s expressible drip was noticeably higher. Surimi gel appears to be stabilized against lipid oxidation, as demonstrated by low PV and TBARS levels, when produced with T4. Because of the low level of TBARS, all 10 panelists rated rancid odor as low (~1 out of 4), with no significant variations across treatments. Only treatments with T4 and T6 tended to have a lower fishy odor score as compared to unwashed mince. Scanning electron microscope demonstrated that surimi gels washed with all washing media exhibited microstructures that were very comparable, with the exception of the T6 treatment, which had big pores and aggregates. Based on the quality features, T4 appeared to be the optimal medium to enhance the gel functionality of mackerel surimi.

## 1. Introduction

Worldwide, customers are becoming more interested about choosing healthy foods. Surimi is primarily made up of myofibrillar protein, which is high in protein, low in fat, and low in cholesterol [1]. Surimi is a Japanese word that means washed fish mince. Basically, surimi is made by continually rinsing mechanically separated fish mince in cold water (5–10 °C) until the majority of the water-soluble components are eliminated. Myofibrillar proteins, which are primarily responsible for the creation of gels, are highly concentrated in the subsequent surimi [2].

Thailand is one of Southeast Asia’s largest surimi manufacturers. Thailand has approximately 16 surimi factories, with a total annual production of about 100,000 metric tons, with 80% shipped to Japan and Korea and the remainder exported to Singapore and other nations [3,4]. In Thailand, surimi is typically produced with lean fish or fish with white meat. Dark-fleshed fish, such mackerel and sardine, have received greater attention as a viable substitute raw material for the manufacturing of surimi due to the limited availability of fish, particularly lean fish [2]. Mackerel, such as *Auxis thazard*, was one of the most frequently caught species of dark-fleshed fish in Thailand [5]; it can also be used to make surimi [5,6]. However, the abundance of dark tissue coupled with high level of lipids and myoglobin presents a challenge in the production of surimi from those dark-fleshed fish [7]. In addition to discoloration, myoglobin-mediated lipid oxidation appears to be a particular issue in surimi prepared from dark-fleshed fish [8]. Fish muscle lipid oxidation, which results in the development of unpleasant flavors, denaturation of proteins, and diminished gelling capability through peroxide generation, may be the main factor limiting fish muscle persistence in storage [9]. According to research done by Keleher et al. [10], washed mince from dark muscle was substantially more susceptible to lipid oxidation than washed mince from white muscle. Myoglobin and lipid oxidation have a close association that may affect how quickly food deteriorates [11]. Myoglobin autoxidation, which contributed significantly to the production of hydroperoxide, stimulated the oxidation of lipids in fish muscle [12]. In both microsomes and liposomes systems, Chan et al. [13] and Yin and Faustman [14] found a strong link between the oxidation of oxymyoglobin and lipids.

Washing is frequently acknowledged as a crucial step in establishing the suitability of surimi and surimi-like materials that resemble it for ingestion [15,16,17,18]. Classical surimi is made by repeatedly washing minced muscles in cold water to remove the majority of the soluble protein. The removal of blood, pigment, connective tissue, off-flavor compounds, and sarcoplasmic proteins, as well as the ultimate rise in the concentration of myofibrillar proteins, are all achieved through washing [15]. To enhance the quality of dark-fleshed fish surimi, several treatments and washing techniques have been widely researched. For the preparation of surimi from fish with dark flesh, salt solution has been utilized as a washing medium. According to Chaijan et al. [2], the breaking force of gels from minced sardine (*Sardinella gibbosa*) and mackerel (*Restrilligar kanagurta*) washed with NaCl solution (0.2% for sardine and 0.5% for mackerel) was stronger than that of gels from mince rinsed with water. The rise in fold test results suggests that washing with a NaCl solution may increase the gel strength of the muscles in cod and flounder. The gel strength of the red hake muscle was not, however, affected by the NaCl solution [19]. According to Chaijan et al. [2], myoglobin was released from the muscle after washing as a result of NaCl’s ability to weaken the connections that hold myoglobin to the structure of the muscle.

Somjid et al. [6] described the use of cold carbonated water (CW) as a washing medium in the processing of mackerel (*A. thazard*) surimi. One-cycle washing with CW, followed by two-cycle washing with cold water, was the most effective for improving the overall quality of surimi gel. This circumstance resulted in surimi with a reduced lipid content, increased gel strength, and low expelled water. From the preliminary study, washing with cold CW in the presence of 0.6% NaCl (*w*/*v*) in the first washing cycle followed by washing with water in the second and the third cycles offered the surimi the greatest gel strength and water retention potential. However, residual myoglobin and lipids still oxidized to some degrees, leading to a lowered whiteness. To overcome this problem, it is necessary to test herein the incorporation of antioxidant in the washing medium. NaCl may help leaching the undesirable substances whereas antioxidants are compounds that are able to retard or inhibit lipid oxidation [2]. In general, antioxidants can postpone, slow, or prevent the development of rancidity or other flavor deterioration caused by oxidation [20]. Sodium tripolyphosphate has been used as a chelating agent to prevent myoglobin oxidation in a model system [21,22]. Furthermore, sodium tripolyphosphate is considerably more efficient than disodium phosphate or monosodium phosphate in reducing lipid oxidation [23]. The water-based phase of chicken breast muscle with active elements such as histidine-containing dipeptides, glutathione, ascorbate, urate, and bilirubin was found to be a potent inhibitor of hemoglobin-mediated lipid oxidation in washed cod muscle [24,25,26]; the effect was thought to be a result of the binding of iron. Sodium erythorbate is a sodium ascorbate stereoisomer that can prevent lipid oxidation by quenching singlet oxygen, donating hydrogen molecules, and working as a reducing agent [27]. Ethylenediaminetetraacetic acid (EDTA) is a water-soluble chelating agent (log *P* = −0.84), which preferentially assembles in an aquatic environment [28]. Phenolic compounds both in the forms of pure compounds and extracts have been discovered to slow lipid oxidation in fish muscle and related food products [29,30,31]. Phenolic compounds have been shown to inhibit hemoglobin- and iron-promoted oxidation in fish model systems [29,32]. The antioxidant activity of phenolic compounds may also be influenced by their reducing capacity or ability to donate electrons, as well as their chelating properties [29]. Gallic acid, one of the prospective phenolic choices, has strong antioxidant bioactivity and can inhibit lipid oxidation [33]. Citric acid is a biobased polycarboxylic acid chelator extracted from citrus fruits, showing antioxidant activity. Citric acid can cross-link via covalent intermolecular di-ester linkages between the carboxyl groups of the citric acid and hydroxyl groups of the target molecules [34]. Chloride salts like CaCl_2_, MgCl_2_, ZnCl_2_, and NH_4_Cl are known to alter the techno-biofunctionality of protein during gelation [35]. Ofstad et al. [36] reported that adding a mixture of CaC1_2_ and MgC1_2_ to the wash water was found to improve the surimi gel functionality. Also, the addition of CaCl_2_ increased the cross-linking reaction of the myosin heavy chain in salted and minced meat [36].

In practice, myoglobin and lipids should be eliminated as much as possible to generate surimi with high oxidative stability and excellent gelling qualities. Washing is a critical step in surimi processing that can be utilized to leach out such components. However, typical cold-water washing is inapplicable to dark-fleshed fish species. To produce superior surimi from fish with dark meat, the washing medium/process should be tailored. As a result, the objective of this study was to highlight a novel method based on soda–saline washing combined with antioxidants to improve the removal efficiency of myoglobin and lipid from mackerel (*A. thazard*) mince and the overall quality of the resulting surimi. Six treatments (T1–T6) were carried out. Unwashed mince (T1) and conventional water washed surimi (T2) were compared with mackerel mince washed with CW in the presence of 0.6% NaCl at a medium to mince ratio of 3:1 (*v*/*w*) without antioxidant (T3) or with the addition of 1.5 mM EDTA plus 0.2% (*w*/*v*) sodium erythorbate and 0.2% sodium tripolyphosphate (T4), 100 mg/L gallic acid (T5), and 5 mM citric acid containing 8 mM calcium chloride (T6).

## 2. Materials and Methods

### 2.1. Fish Raw Material

Mackerel (*A. thazard*) with an average weight of 90–100 g were bought from Thasala market, Nakhon Si Thammarat, Thailand. The fish were unloaded around 12 h after catch and packed in ice using a fish/ice ratio of 1:2 (*w*/*w*) before being taken within 15 min to the School of Agricultural Technology and Food Industry, Walailak University. The fish were then headed, eviscerated, washed, filleted, and skinned, respectively. Then, the fish meat was uniformly minced using a meat grinder (a 4 mm hole diameter; Panasonic MK-G20MR, Kadoma, Japan). During the time of preparation, the muscles were kept on ice.

### 2.2. Combined Effect of CW, NaCl, and Antioxidant Washing on Myoglobin and Lipid Removal Efficacies, Biochemical Properties, and Gel Functionalities of Mackerel Surimi

To study the influence of CW–saline washing with antioxidants on quality of mackerel surimi, fresh whole mackerel mince was washed with CW in the presence of 0.6% NaCl (optimal NaCl concentration from trial experiment). Commercial CW (Chang^®^, Cosmos Brewery Co. Ltd., Bangkok, Thailand) was used. The treatments were (i) 1.5 mM EDTA plus 0.2% (*w*/*v*) sodium erythorbate and 0.2% sodium tripolyphosphate [37], (ii) 100 mg/L of gallic acid [38], and (iii) 5 mM citric acid containing 8 mM CaCl_2_ [39], using a medium to mince ratio of 3 to 1 (*v*/*w*). The antioxidants were added in the washing medium at the first washing cycle. Conventional washing with cold water and unwashed mince were used as controls. The washing media used in each treatment are listed in Table 1. In a cold environment (4 °C), the mixture was vigorously agitated for 10 min before the washed mince was passed through a layer of nylon screen. Hydraulic press dewatering was used, with a final moisture content of 80%. The weight of the raw fish material was used to compute the yield of surimi from all washing operations. All samples were mixed thoroughly with 4% sucrose and 4% sorbitol before being frozen in an air-blast freezer (Polar DN494 367 Blast Freezer, Campbell town, NSW, Australia) at −21 °C for 1 h. Before being used for analysis, the frozen samples were stored at −18 ± 2 °C. The frozen storage period was no more than one month. Table 1 summarizes the washing media used in each treatment in this study.

#### 2.2.1. Determination of Moisture Content and pH

Unwashed mince and surimi were measured for moisture content and pH using the AOAC [40] and Benjakul et al. [41] methods, respectively.

#### 2.2.2. Determination of Reactive Sulfhydryl (SH) Content and Ca^2+^-ATPase Activity

Using the methods of Ellman [42] and Benjakul et al. [41], the reactive SH content and Ca^2+^-ATPase activity of the natural actomyosin (NAM) isolated from unwashed mince and surimi were determined, respectively.

#### 2.2.3. Determination of Protein Surface Hydrophobicity

According to the procedure described by Chelh et al. [43], the hydrophobicity of nonsolubilized myofibrils was assessed using bromophenol blue (BPB) for electrophoresis. Two hundred microliters of 1 mg/mL BPB (in distilled water) was added to 1 mL of myofibril suspension made in accordance with the technique outlined by Martinaud et al. [44] and thoroughly mixed. A mixture consisting of 200 μL of 1 mg/mL BPB (in distilled water) to 1 mL of 20 mM phosphate buffer (pH 6) served as the control, which was devoid of myofibrils. After being stirred up for 10 min at ambient temperature (28–30 °C) with the controls, the samples were centrifuged at 2000× *g* for 15 min. At 595 nm, the absorbance of the diluted 1/10 supernatant was measured in comparison to a phosphate buffer blank. The amount of BPB bound indicated by the following formula was given as the surface hydrophobicity:(1)$$\mathrm{BPB}\;\mathrm{bound}\;(µg)=200\;µg\;\times \;\left[\frac{A_{control}-A_{sample}}{A_{control}}\right]$$
where A = absorbance at 595 nm.

#### 2.2.4. Lipid Extraction and Lipid Oxidation Indices

By using the Bligh and Dyer method [45], the lipid was extracted. According to the methods of Panpipat et al. [46] and Frankel et al. [47], the peroxide value (PV) and conjugated diene (CD) of the lipid sample were determined. The ground sample was used to carry out the thiobarbituric acid reactive substances (TBARS) assay according to Buege and Aust’s protocol [48].

#### 2.2.5. Determination of Myoglobin Content and Metmyoglobin Content

The extractable myoglobin concentration was assessed spectrophotometrically at 525 nm using Benjakul and Bauer’s technique [49]. Myoglobin content was determined using a millimolar extinction coefficient of 7.6 and a molecular mass of 16,110 [50].

The absorbance values were also read at 503, 525, 550, 557, 582, and 630 nm and the metmyoglobin content was estimated by modified Krzywicki’s equations [51] as followed:Metmyoglobin (%) = −0.159R_1_ − 0.085R_2_ + 1.262R_3_ − 0.520(2)
where R_1_ = A_582_/A_525_, R_2_ = A_557_/A_525_, and R_3_ = A_503_/A_525_.

#### 2.2.6. Determination of Heme Iron and Non-Heme Iron 

The heme iron content was assessed using the method provided by Benjakul and Bauer [49]. Heme iron was determined based on myoglobin, which contains 0.35% iron [50]. The non-heme iron content was measured using the method proposed by Schricker et al. [52].

#### 2.2.7. Determination of Color

*L** (lightness), *a** (redness/greenness), and *b** (yellowness/blueness) values were measured in triplicate using a colorimeter (Hunter Assoc. Laboratory; Reston, VA, USA).

#### 2.2.8. Determination of Rheological Properties

The rheological parameters (elastic modulus G′, viscous modulus G″, and tan δ) of unwashed mince and surimi pastes were tested using a HAAKE MARS 60 Rheometer (Thermo Fisher Scientific Inc., Yokohama, Japan) from 10 to 90 °C at a rate of 2 °C per min, as described by Somjid et al. [4].

### 2.3. Gel Preparation and Analyses

Frozen surimi or unwashed mince samples were thawed at 4 °C until the center’s temperature approached 0 °C. After that, the samples were sliced into little pieces and the moisture level was set to 80%. To produce a homogenous sol, dry NaCl (2.5% *w*/*w*) was added to the samples and chopped for 5 min. The sol was subsequently placed into a 2.5 cm diameter polyvinylidine casing and both ends were firmly sealed. The sample was then allowed to incubate at 40 °C for 30 min before being heated at 90 °C for 20 min. Prior to analysis, the gels were cooled in ice water then left at 4 °C for 24 h [53].

The gels’ textures (breaking force and deformation) were analyzed with a TA-XT2 texture analyzer (Stable Micro Systems, Godalming, Surrey, UK), and gel strength was calculated by multiplying the breaking force and deformation [53]. The expressible moisture of gels was determined using the method of Phetsang et al. [54] and reported as a percentage of sample mass. Colorimetric values of gels were obtained and the whiteness was calculated as follows:Whiteness = 100 − [(100 − *L*^∗^)^2^ + *a*^∗2^ + *b*^∗2^]^1/2^(3)

PV and TBARS of gels were measured using the methods described previously.

Ten trained panelists (5 females and 5 males) with ages between 23 to 35 years from the laboratory staff and graduate students of the Food Technology and Innovation Research Center of Excellence, School of Agricultural Technology and Food Industry, Walailak University, performed sensory examination of gel. The panelists were regular surimi consumers with no surimi allergies. The panelists were chosen based on their sensitivity to fishy and rancid odors and their extensive expertise in assessing the off-odors and off-flavors of cooked seafood and surimi gel. All of them were trained to have rich professional knowledge on fishy and rancid odor [4]. Gel samples were cut to a thickness of 1 cm and a diameter of 2.5 cm, equilibrated at room temperature for 30 min, and coded with 3-digit random numbers. Before the sensory evaluation, samples were kept in a plastic cup with a cover. After opening the cup, the panelists were instructed to sniff the headspace above the samples. The intensity of fishy and rancid odors was graded on a 5-point scale varying between none (score = 0) through strong (score = 4) [54]. The Walailak University Human Research Ethics Committee (WUEC-21-125-02) authorized the study’s protocol.

Gel microstructures were examined with a scanning electron microscope (SEM) (GeminiSEM, Carl Ziess Microscopy, Jena, Germany) at a 10 kV accelerating voltage [4].

### 2.4. Statistical Analysis

Throughout this study, a completely randomized design was adopted. All experiments were carried out in triplicate (*n* = 3) except for the off-odor score which was evaluated by 10 trained panelists. The data were reported by mean ± standard deviation. Data were subjected to ANOVA and mean comparisons were performed using Duncan’s Multiple Range Test. The statistical program SPSS was used for the statistical analysis.

## 3. Results and Discussion

### 3.1. Yield and Biochemical Properties

In general, the mackerel surimi yield was impacted by the washing media. Cold carbonated water with or without antioxidants had an equivalent or superior performance when compared to conventional water washing. On average, based on the weight of the raw mince, the yields of all treatments were around 75–83%. T3 and T5 had higher yield than conventional washing (T2), T4, and T6 (Table 2).

Table 3 shows the effect of CW–saline washing with antioxidants on biochemical features of mackerel surimi in comparison with unwashed mince. The pH of the unwashed mince was 5.58 and it rose to a range of 5.99–6.46 after washing with T2–T5 (*p* < 0.05). The pH of conventional surimi (T2) was 6.16. The elimination of acidic elements from the mince or the impact of the media’s pH both contributed to a rise in pH following washing with T2–T5. Due to the presence of citric acid in the washing medium, only T6 noticed a pH reduction to 5.47 (*p* < 0.05) (Table 3).

All of the surimi prepared with T2–T6 had higher reactive SH content than unwashed mince (T1) (*p* < 0.05) despite the fact that the reactive SH concentration varied between treatments (Table 3). This resulted from the SH content being exposed during washing, which suggested that proteins may have been denaturing or unfolding, making them more likely to undergo thermal gelation in the following stage [4,5,6]. The surimi produced by T2 and T4 showed the greatest reactive SH content (*p* < 0.05), and those produced by T3/T5 and T6 were next in line (Table 3). The SH level of frigate mackerel surimi made by ultrasound-aided washing was around 6 mol/10^8^ g protein [4,55], which was near to the value reported herein, but higher SH content can be found in surimi-like material from spent duck meat [16] and goat meat [18].

Interestingly, the Ca^2+^-ATPase activity was significantly lower in all the surimi than in the control (Table 3). The Ca^2+^-ATPase activity in the surimi was 1.51–1.86 μmolPi/mg protein/min as opposed to the control’s 6.45 μmolPi/mg protein/min. The Ca^2+^-ATPase activity of conventional surimi from the same fish species was reported to be 1.79–2.10 μmolPi/mg protein/min [4,55], which was within the range of the value found in this study. Comparing T3–T6 with conventionally washed surimi (T2), T3–T6 tended to have higher residual Ca^2+^-ATPase activity. When compared to unwashed mince, Das et al. [56] discovered a steady decrease in Ca^2+^-ATPase activity in mackerel, sardine, croaker, and pink perch surimi. Although the Ca^2+^-ATPase activity was associated with the integrity of myosin, the denaturation that occurs during washing may be the cause of the decrease in this enzyme’s activity. Theoretically, it might potentially result in a decrease in the gelation functionality but proper unfolding, such as partial denaturation, may have a positive impact on gel-forming ability due to the exposure of some reactive groups, such as SH and hydrophobic regions, which facilitate the protein–protein interaction even though it loses its integrity as demonstrated by lowered Ca^2+^-ATPase [4].

All the surimi had more hydrophobic surfaces than unwashed mince (*p* < 0.05; Table 3). This supported the idea that proteins unfolded during washing because the internal hydrophobic moieties or patches were made visible on the protein surface [57]. Except for T6, which had the highest surface hydrophobicity and thus the largest degree of denaturation produced by the highest acidity in this treatment, washing without antioxidant (T2) resulted in more protein unfolding and loss of the myosin integrity than did washing with antioxidant (T3–T5). The surface hydrophobicity of mackerel surimi produced by single washing was reported as 41.42 μg of BPB bound [4], which was roughly comparable to the values observed in this study.

### 3.2. Lipid Content and Lipid Oxidation

In comparison to unwashed mince, Table 4 displays the effects of CW–saline washing with antioxidants on the lipid content and lipid oxidation indices of mackerel surimi. After washing, lipids dramatically decreased from 1.44 g/100 g to roughly 0.35–0.51 g/100 g, accounting for a 65–76% reduction. The lipid reduction was within the range of the manufacture of surimi-like material from duck meat using three-cycle washing with different media (tap water, NaCl, sodium bicarbonate, and sodium phosphate buffer), which eliminated the lipids between 62.5 and 78.6% [17].

The lowest residual lipid concentration was seen in T5 containing gallic acid, which was likely caused by phenolic–lipid conjugation and subsequent elimination during dewatering. In water-based environments, lipids self-construct into lipid bilayers, a two-dimensional biomolecular sheet composed of two layers of lipid molecules oriented almost parallel to create a hydrocarbon center and hydrophilic headgroups deposited on each side of the hydrocarbon center [58]. According to Karonen [58], depending on their structures, concentrations, lipid compositions, and environmental circumstances, polyphenols can interact with and enter lipid bilayers in a variety of ways. The structure and biophysical characteristics of the lipid bilayer can be disturbed and altered when polyphenol penetrates it. Despite claims that T6’s components, 5 mM citric acid plus 8 mM CaCl_2_, made it easier to remove membrane lipids, the residual lipid content was the same among the surimi [39]. It might be because mackerel mince has more neutral lipids, which these compounds find difficult to extract. As previously mentioned, these lipids may interact with polyphenol more frequently and be more effectively removed by T5 as a result.

The lipid oxidation in unwashed mince and surimi produced by different washing media was monitored using PV, CD, and TBARS (Table 4). From the results, washing can enhance lipid oxidation in some treatments of minced fish due to some reasons. It has been reported that washing disrupts the initial pro-oxidant–antioxidant balance in muscle tissue, contributing to the observed discrepancies in the oxidative condition of the products produced [38]. Although the unwashed mince had the highest level of lipids, the surimi did not have the highest incidence of lipid oxidation. The typical washing (T2) without antioxidant had the greatest PV value (*p* < 0.05), followed by T6, T1/T4/T5, and T3. Unwashed mince (T1) had the lowest value for CD, followed by T3, T6, T2, T4, and T5 (*p* < 0.05). CD production appeared to be enhanced by washing both with and without antioxidants. Interestingly, T3 had the lowest PV and CD even though it only included 0.6% NaCl without antioxidant. Its mechanism is still unknown; however, it might be because of this treatment’s elimination of pro-oxidants and stabilization of endogenous antioxidants. Additionally, it could be influenced by the rate at which peroxide builds up and transforms into advanced compounds. PV or CD alone might not be sufficient to assess the effectiveness of the treatment. TBARS, a secondary lipid oxidation product, was therefore examined. The group with the highest TBARS value according to the results was T3; hence, this treatment may promote the oxidation of the lipid rather than stabilize it. Hydroperoxide is a main oxidation product during fish preservation that easily decomposes to a variety of volatile chemicals such as aldehydes, ketones, and alcohols [59]. The generation of secondary lipid oxidation products is one of the primary causes of the formation of unpleasant odors in fish muscle. PV and CD measurement are the two most popular ways to identify principal oxidation products. Malondialdehyde, a byproduct of secondary oxidation, is the most common lipid oxidation biomarker that can be measured using the TBARS method [60]. T1/T4 appeared to have the lowest TBARS value, followed by T2/T6 and T3/T5. Unwashed mince and certain antioxidants were found to be able to stabilize lipid against oxidation as well as to prevent undesirable secondary oxidation products. T4 was one of the treatments that tended to be the most effective at stabilizing the lipids in mackerel surimi. T4 is made of CW + 0.6% NaCl in the presence of 1.5 mM EDTA, 0.2% sodium erythorbate, and 0.2% sodium tripolyphosphate. In the aqueous phase, sodium erythorbate can also function as a free radical scavenger. Non-heme iron would remain in the active reduced ferrous form if heme iron was reduced using a reducing agent like sodium erythorbate. Then, chelating agents for such reduced iron included EDTA and sodium tripolyphosphate. Additionally, sodium tripolyphosphate serves purposes other than antioxidation, like enhancing proteins’ ability to bind to water [10,61]. Only the first wash was supplemented with sodium tripolyphosphate in this investigation. When the muscle tissues were introduced in the second and third washes, they swelled to the extent that the material could not be dewatered [10]. According to Larouche et al. [62], lipids are categorized as not oxidized (TBARS value < 1.5 mg MDA/kg), moderately oxidized (1.6 < TBARS value < 3.6), or oxidized (TBARS value > 3.7). Thus, all of the surimi in this study were within the permissible requirements for the TBARS level.

### 3.3. Myoglobin and Its Related Species

Table 5 compares the myoglobin, metmyoglobin, heme iron, and non-heme iron contents of mackerel surimi and unwashed mince after washing with cold carbonated water–saline washing with antioxidants. Unwashed mince (T1) had a myoglobin level of roughly 44 mg/100 g, which decreased after washing with T2 to T6 (*p* < 0.05). The heme iron content followed the same pattern. T2/T4 had the lowest residual myoglobin/heme iron level, followed by T3 and T5/T6. Depending on the type of raw materials used and the washing techniques employed, it was reported that surimi had lower myoglobin, heme iron, and non-heme iron levels than unwashed mince [4,16,17,18,51,55].

When 1.5 mM EDTA, 0.2% sodium erythorbate, and 0.2% sodium tripolyphosphate were added, the myoglobin and heme iron concentrations tended to be similar to the treatment washed with cold water (T2). However, the combination of CW and NaCl did not improve the myoglobin removal capability. Antioxidant-treated T4/T5/T6 exhibited a lower non-heme iron level than untreated T1/T2/T3 did, though. The surimi with the least non-heme iron concentration seems to be produced by T5 with gallic acid. According to various studies [63,64,65], the release of non-heme iron and the formation of metmyoglobin may be prevented by the complexation of myoglobin/heme protein with phenolic compounds. Due to the chelators’ presence, T4 with phosphate and EDTA, and T6 with citric acid may each prevent the release of non-heme iron [20].

Unwashed mince had the highest concentration of metmyoglobin development and T5 had the lowest. The metmyoglobin content was the same in the other treatments (T2, T3, T4, and T6). Regardless of the washing medium, washing can increase the redox stability of myoglobin, as shown by the relatively low metmyoglobin levels in surimi when compared to unwashed mince. In this investigation, the metmyoglobin content of all surimi washed with antioxidant-infused CW–saline solutions was lower than that of frigate mackerel washed conventionally, as reported by Somjid et al. [4]. The remaining myoglobin and its species, however, could alter once more during thermal gelation, which would then affect the gel’s characteristics (see below).

### 3.4. Color

In comparison to unwashed mince, Table 6 displays the color parameters (*L**, *a**, and *b** values) of mackerel surimi. Surimi typically had lighter color than unwashed mince, which was demonstrated by a higher *L** value. After washing, the colors were eliminated, and the water’s ability to hold on to the muscle tissue made the surimi lighter. After washing, redness (*a**) was similarly lessened (*p* < 0.05); however, the *a** value varied between treatments. In comparison to T5 and T6, T2, T3, and T4 produced a surimi with a lower *a** value. This was connected to the remaining myoglobin and heme iron levels (Table 5). Surimi had a *b** value that was equal with or higher than that of unwashed mince, where all samples’ *b** values ranged from 4.6 to 5.7. Surimi’s color may vary depending on the stability of the pigment as well as any remaining pigment in the muscle. The color of the surimi may also be impacted by color reactions, such as the browning reaction after washing [5]. The effect of heat treatment-induced pigment oxidation or the creation of color-related compounds, which in turn affects the net whiteness of the final surimi gel, caused the color of surimi to change once more after thermal gelation (see below).

### 3.5. Rheology

Rheology is a tool utilized to determine the physicochemical elements that contribute to gelation, which is the foundation for texture creation [66]. Figure 1 depicts the dynamic viscoelastic behavior of unwashed mackerel mince and surimi generated through altering the washing media across the temperature shift from sol to gel. The storage modulus (G′) was used for estimating gel formation (Figure 1a). A rise in G′ suggested that the rigidity of the sample had increased due to the occurrence of an elastic structure [67]. With the exception of T6, which was decreased from the start until roughly 42 °C and then constantly lifted, the G′ in all samples was steadily increased from approximately 35 °C to around 45 °C, as depicted in Figure 1a. The significantly higher G′ of T6 at the start showed protein aggregation in this surimi. The “gel setting” stage, which can be seen in T1–T5, was used to create the first protein network structure. At this stage, myosin will unravel in order to enable ordered polymerization and the initial elasticity of proteins would be gone [68]. According to Buamard et al. [69], this spectrum includes the formation of protein connectivity via weak links among the molecules of proteins, such as hydrogen bonds. The setting phenomenon may be minimized in T6, which has been prepared with a solution containing citric acid, resulting in greater protein denaturation as demonstrated by the maximum surface hydrophobicity (Table 3). The G′ was subsequently increased again, culminating at roughly 60–65 °C, indicating the formation of a robust gel network due to stronger attractive interactions (e.g., disulfide and hydrophobic linkages). Following that, G′ progressively increased until it reached its peak at the end. In the “gel strengthening stage”, myosin aggregation and initially cross-linking were created. T1 and T4 had the same G′ pattern, while T2, T3, and T5 likewise had the same G′ pattern.

Unwashed mince and surimi viscosity modulus (G″) curves were frequently similar to G′ (Figure 1b). The G″ followed the same trend as the G′ (Figure 1a), but the degree of organized interconnection was substantially greater, leading to a lower G″ value, indicating that the most crucial component in the emergence of surimi gel was its elastic composition. The tan δ of all samples was less than 1.0 because G′ values were greater than G″ values (Figure 1c). Therefore, all of the samples exhibited the characteristics of an elastic fluid with better gelation properties [70]. The tan δ was about the same even though the final G′ and G″ of the surimi varied, suggesting that all surimi will go through a similar amount of sol–gel transformation after heat gelation. As a result, all of the samples can gel at different degrees throughout the two-step heating (40 °C/30 min → 90 °C/20 min). The rheological properties of all surimi washed with antioxidant-infused CW–saline solutions were similar to those of other dark-fleshed fish surimi; however, the values of G′, G″, and tan δ differed to some extent [4].

### 3.6. Textural Properties and Color of Gel

Table 7 compares the effects of cold carbonated water–saline washing with antioxidants on the textural characteristics, whiteness, and expressible drip of mackerel surimi gel to unwashed mince gel. Unwashed mince (T1) (*p* < 0.05) had the lowest breaking force, deformation, and corresponding gel strength. Those textural characteristics were enhanced after washing to create surimi. T6/T5 were greater than T4, T2, and T3, consequently, in terms of breaking force. T4/T5 were greater than T2/T3/T6 for deformation. Surimi created by T4/T5 had the highest gel strength when taking into account both breaking and deformation, followed by T2/T6, T3, and unwashed mince (T1). Although T4 and T5 had identical gel strengths, T5’s expressible drip was noticeably higher than T4’s, indicating that agglutination rather than gelation may have taken place in T5 due to the latter’s higher breaking force and expressible drip. The same pattern was seen in T6, which may be explained by an unbalanced relationship between protein–protein and protein–water interactions. T6 had the greatest expressible drip rate overall, followed by T1, T5, T3, T2, and T4. T4 therefore appeared to be the best medium to improve the gel functionality of mackerel surimi based on the textural characteristics. The textural characteristics of all the surimi washed with antioxidant-infused CW–saline solutions were relatively similar to those of other dark-fleshed fish surimi but were nevertheless lower than those of other animal surimi-like materials [4,16,17,18].

Unwashed mince (T1) had the lowest value for whiteness because it had the largest myoglobin and heme iron contents (Table 5), as well as lower *L** and higher *a** values (Table 6). Because T5 and T6 had the largest residual myoglobin content, the surimi gel, in particular T2, T3, and T4, tended to be whiter than T5 and T6 (Table 5). Lower whiteness values may result from these treatments because heating causes myoglobin to oxidize and transform into metmyoglobin. Other processes that may contribute to a decreased whiteness of unwashed mince gel include lipid oxidation and the Maillard browning reaction [6]. The darker color of the protein isolate gel may have been caused by the Maillard reaction products and brown metmyoglobin [54]. The appearances of unwashed mince and surimi gels are depicted in Figure 2.

### 3.7. Lipid Oxidation and Sensorial Property of Gel

Table 8 shows the effect of CW–saline washing with antioxidants on lipid oxidation and fishy odor score of mackerel surimi gel in comparison with unwashed mince gel. The PV of all gels was higher than their initial surimi counterparts except for T4 (Table 4 and Table 8). Unwashed mince gel had the highest PV, followed by gel of surimi prepared by T6, T2/T5, T3, and T4. T4 was the most effective antioxidant treatment against the development of primary lipid oxidation products during thermal gelation. Other antioxidant treatments (T5 and T6) did not prevent peroxide development. According to Ghimire et al. [71], the PV threshold limit in beef meatballs during refrigerated storage was set at 1 meq peroxide/kg fat. T4 was considered an effective medium in retarding the development of primary lipid oxidation product in mackerel surimi due to that permitted level of PV. All of the gels had slightly varied TBARS values between treatments, both before and after thermal gelation. T5 with gallic acid, and T6 with citric acid and calcium chloride had greater TBARS values than the other treatments, including the unwashed mince gel. TBARS is a measurement of aldehydic lipid oxidation product that can be produced and volatilized during heating, influencing the TBARS value [54]. The TBARS level in all of the surimi gel in the present study was within the permissible range (TBARS < 1.5 mg MDA/kg) [62]. In terms of quality, the gel from surimi, particularly that made using T4, appeared to be stabilized against lipid oxidation, as indicated by low PV and TBARS levels. Furthermore, the carry through effect, which refers to antioxidants’ ability to tolerate thermal treatments [72], could be one element impacting surimi gel lipid oxidation.

Since mackerel has dark-colored flesh, lipid oxidation caused by heme iron can be one of the detrimental effects. T4 contains 0.2% sodium erythorbate, 0.5% sodium tripolyphosphate, and 1.5 mM EDTA. Based on reports, EDTA and STPP are the iron chelators that are most effective at postponing hemoglobin-mediated lipid oxidation. According to Maestre et al. [73], EDTA has a significant capability of forming complexes with ferrous ions because 10 μM EDTA may chelate up to 12.9 μM ferrous iron. In fish muscles supplemented with STPP and EDTA, the production rates of TBARS also showed decreased lipid oxidation propagation and higher fish redness. According to Maestre et al. [73], citric acid was unable to impede the production of lipid oxidation products such as lipid peroxides and TBARS as well as the rapid loss of redness. According to reports, EDTA is a significantly more effective metal chelator than citric acid, making it far better at removing trace metal contaminants [74].

It has been noted that certain prominent antioxidants exhibit pro-oxidant activity. At least three factors can influence an antioxidant’s function, converting it to a pro-oxidant; these factors include the presence of metal ions, the antioxidant’s concentration in matrix settings, and its redox potential [75]. Phenolics can also promote oxidation, particularly when redox-active metals are present. The redox process of these substances is catalyzed by the presence of iron or copper, which can lead to the production of phenolic radicals that can damage DNA and lipids [75,76].

Because of the low level of TBARS, all of the panelists judged rancid odor as low (≤1 out of 4) with no significant differences across treatments. The fishy odor score, on the other hand, differed between treatments. Unwashed mince (T1) had the greatest fishy odor score, whereas surimi had a reduced fishy odor score due to the elimination of volatile compounds, blood, and other odorants in fish muscle, which was modified by washing media [4]. Certain substances may be discovered in the muscle postmortem, while others may be produced during the pre-handling and washing steps. Only T4 and T6 tended to have a lower fishy odor score as compared to unwashed mince. T4 washing can thus effectively postpone the production of off-odors and lipid oxidation.

### 3.8. Microstructure of Gel

Figure 3 depicts the microstructures of unwashed mince gel and surimi gels generated using various washing methods. The network structure present in each sample of gel demonstrated the elastic nature of the gels. Mince gel (T1) that had not been washed exhibited porous networks with a lot of microscopic holes that were dispersed equally. It was also observed that several fibrous proteins cross-linked, creating a spongy structure with tiny fibrous networks. This was because unwashed mince contained a variety of substances, particularly sarcoplasmic proteins and lipids, both of which had a limited ability to gel. According to Arfat and Benjakul [77], sarcoplasmic proteins are able to attach to myofibrillar proteins and prevent the development of a strong gel network. It has been proven that residual lipids reduce the gel strength of tropical fish surimi. Lipids probably disrupted the bond between protein molecules since they are unable to create a matrix of gel [2]. Surimi gels (T2–T6) washed with various washing media, on the other hand, showed relatively comparable microstructures, with the exception of T6, which had large pores and aggregates. Because of the influence of encapsulated capillary water, myosin gel networks formed some concentrated dense regions and distributed pores of various size and depth, resulting in improved strength and some larger size, deep, and localized holes as seen in T2, T3, and T4. It is apparent that surimi has a greater ability for water retention (Table 7).

## 4. Conclusions

The incorporation of antioxidant, particularly 1.5 mM EDTA + 0.2% sodium erythorbate + 0.2% sodium tripolyphosphate) in CW containing 0.6% NaCl, can be a strategy in improving the gel-forming ability and oxidative stability of mackerel (*A. thazard*) surimi. This medium can be referred to as the lipid-stabilizing technique, which was created to reduce oxidation and rancidity while also improving gel functionality in dark-fleshed fish. The findings from this study will aid in establishing the feasibility of mackerel, a dark-flesh fish species, as a raw material for surimi manufacturing, which will benefit the surimi industry’s sustainability. Further study can be conducted to investigate the storage stability of surimi made using this washing medium in order to increase its industrial application.

## Figures and Tables

**Figure 1 foods-12-03178-f001:**
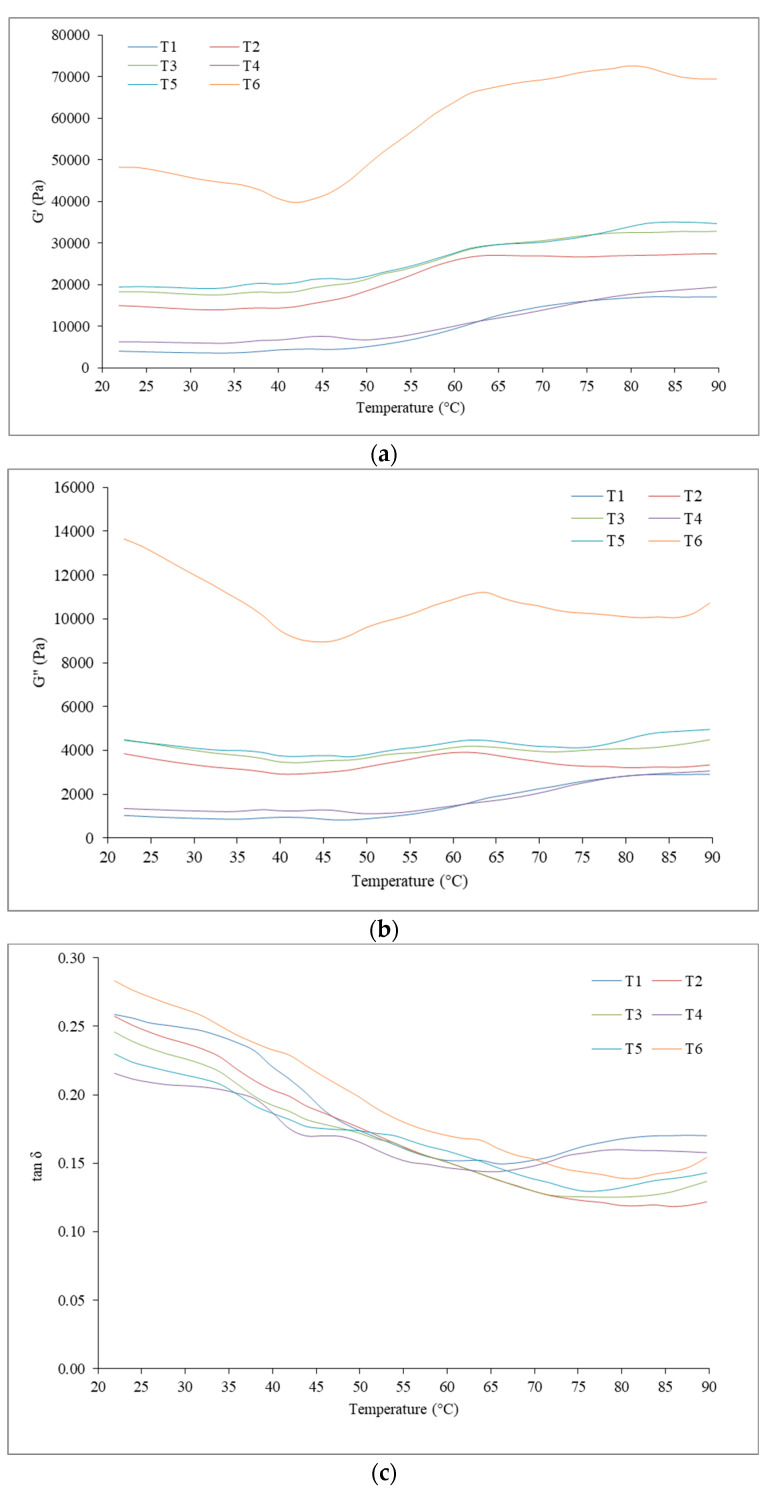
Effect of antioxidant-infused soda–saline washing on rheological properties including G′ (**a**), G″ (**b**), and tan δ (**c**) of mackerel (*A. thazard*) surimi in comparison with unwashed mince. T1–T6 details can be seen in Table 1.

**Figure 2 foods-12-03178-f002:**
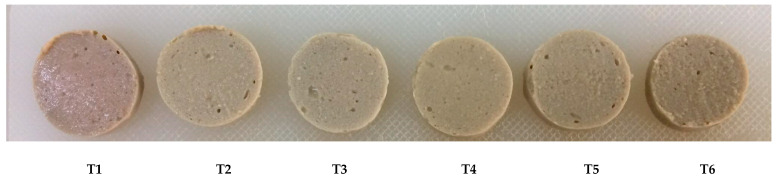
Appearance of mackerel (*A. thazard*) surimi gels as affected by antioxidant-infused soda–saline washing. T1–T6 details can be seen in Table 1.

**Figure 3 foods-12-03178-f003:**
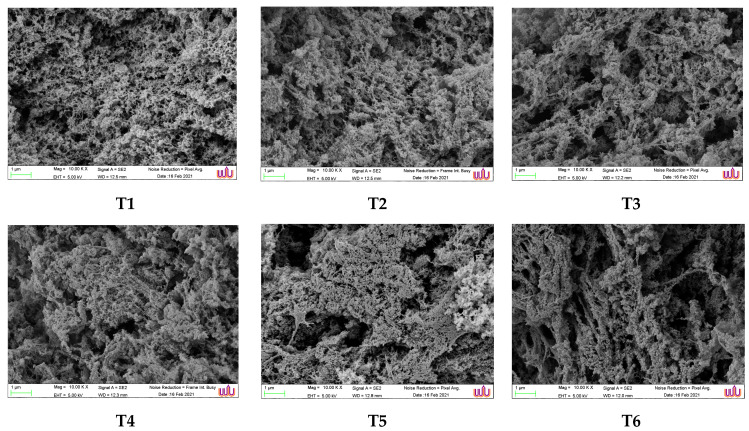
Microstructure of gels from mackerel (*A. thazard*) surimi as affected by antioxidant-infused soda–saline washing (magnification: 10,000 × EHT: 5.0 kV). T1–T6 details can be seen in Table 1.

**Table 1 foods-12-03178-t001:** Washing media used in each treatment.

Treatment	1st Cycle	2nd Cycle	3rd Cycle
T1 (unwashed mince)	-	-	-
T2 (conventional washing)	water	water	water
T3	CW + 0.6% NaCl	water	water
T4	CW + 0.6% NaCl + 1.5 mM EDTA + 0.2% sodium erythorbate + 0.2% sodium tripolyphosphate	water	water
T5	CW + 0.6% NaCl + 100 mg/L gallic acid	water	water
T6	CW + 0.6% NaCl + 5 mM citric acid + 8 mM CaCl_2_	water	water

CW = cold carbonated water.

**Table 2 foods-12-03178-t002:** Effect of antioxidant-infused soda–saline washing on yield of mackerel (*A. thazard*) surimi.

Treatment	Yield (%) *
T1 (unwashed mince)	100.00 ± 0.00 a
T2	76.17 ± 1.60 c
T3	82.68 ± 1.10 b
T4	75.88 ± 1.30 c
T5	81.16 ± 1.52 b
T6	75.55 ± 1.20 c

* The values are reported as the mean ± standard deviation of three determinations. Significant differences (*p* < 0.05) are indicated by different letters. T1–T6 details can be seen in Table 1.

**Table 3 foods-12-03178-t003:** Effect of antioxidant-infused soda–saline washing on biochemical features of mackerel (*A. thazard*) surimi in comparison with unwashed mince.

Treatment	pH	Reactive Sulfhydryl Content (mol/10^8^ g Protein)	Ca^2+^-ATPase Activity (μmolPi/mg Protein/min)	Surface Hydrophobicity; BPB Bound (μg)
T1	5.58 ± 0.01 e	3.26 ± 0.21 d	6.45 ± 0.20 a	27.46 ± 0.55 e
T2	6.16 ± 0.01 c	4.36 ± 0.07 a	1.51 ± 0.03 c	38.76 ± 0.93 b
T3	6.22 ± 0.01 b	3.88 ± 0.06 b	1.68 ± 0.06 bc	36.85 ± 0.82 c
T4	6.46 ± 0.01 a	4.18 ± 0.18 a	1.86 ± 0.03 b	36.35 ± 0.31 cd
T5	5.99 ± 0.02 d	3.87 ± 0.13 b	1.67 ± 0.03 bc	35.53 ± 0.95 d
T6	5.47 ± 0.01 f	3.55 ± 0.07 c	1.68 ± 0.03 bc	44.51 ± 0.43 a

The values are reported as the mean ± standard deviation of three determinations. Significant differences (*p* < 0.05) are indicated by different letters in the same column. T1–T6 details can be seen in Table 1.

**Table 4 foods-12-03178-t004:** Effect of antioxidant-infused soda–saline washing on lipid content and lipid oxidation indices of mackerel (*A. thazard*) surimi in comparison with unwashed mince.

Treatment	Lipid (g/100 g)	Peroxide Value (meq/kg Lipid)	Conjugated Diene	Thiobarbituric Acid Reactive Substances (mg MDA Equivalent/kg Sample)
T1	1.44 ± 0.01 a	1.45 ± 0.11 c	34.47 ± 0.26 f	0.14 ± 0.04 c
T2	0.40 ± 0.04 b	4.08 ± 0.27 a	76.86 ± 0.65 c	0.27 ± 0.05 ab
T3	0.45 ± 0.04 b	0.42 ± 0.08 d	58.40 ± 0.66 e	0.32 ± 0.05 a
T4	0.45 ± 0.05 b	1.61 ± 0.20 c	81.66 ± 0.38 b	0.21 ± 0.05 bc
T5	0.35 ± 0.01 c	1.55 ± 0.01 c	92.25 ± 0.20 a	0.34 ± 0.07 a
T6	0.51 ± 0.05 b	2.34 ± 0.11 b	67.23 ± 0.44 d	0.24 ± 0.05 ab

Values are given as mean ± standard deviation from triplicate determinations. Different letters in the same column indicate significant differences (*p* < 0.05). MDA = malondialdehyde. T1–T6 details can be seen in Table 1.

**Table 5 foods-12-03178-t005:** Effect of antioxidant-infused soda–saline washing on myoglobin content, metmyoglobin content, heme iron content, and non-heme iron content of mackerel (*A. thazard*) surimi in comparison with unwashed mince.

Treatment	Myoglobin (mg/100 g Sample)	Heme Iron (mg/100 g Sample)	Non-Heme Iron (mg/g Sample)	Metmyoglobin (%)
T1	44.01 ± 0.51 a	125.75 ± 1.46 a	13.45 ± 0.00 a	26.23 ± 1.08 a
T2	4.83 ± 0.77 d	13.79 ± 2.20 d	12.88 ± 0.00 a	6.13 ± 0.12 b
T3	7.98 ± 0.85 c	22.81 ± 2.41 c	13.55 ± 0.00 a	5.60 ± 0.14 b
T4	6.42 ± 1.10 cd	18.33 ± 3.14 cd	11.73 ± 0.00 b	5.71 ± 0.12 b
T5	15.21 ± 1.66 b	43.14 ± 4.20 b	10.70 ± 0.00 c	3.40 ± 2.38 c
T6	14.05 ± 0.52 b	40.18 ± 1.45 b	11.73 ± 0.00 b	5.43 ± 0.31 b

The values are reported as the mean ± standard deviation of three determinations. Significant differences (*p* < 0.05) are indicated by different letters in the same column. T1–T6 details can be seen in Table 1.

**Table 6 foods-12-03178-t006:** Effect of antioxidant-infused soda–saline washing on color of mackerel (*A. thazard*) surimi in comparison with unwashed mince.

Treatment	*L**	*a**	*b**
T1	16.73 ± 0.16 b	2.15 ± 0.33 a	4.89 ± 0.34 bc
T2	18.03 ± 0.40 a	0.98 ± 0.10 bc	5.68 ± 0.27 a
T3	17.91 ± 0.10 a	0.62 ± 0.38 d	4.57 ± 0.22 c
T4	18.16 ± 0.36 a	0.84 ± 0.12 cd	5.70 ± 0.11 a
T5	17.61 ± 0.12 a	1.26 ± 0.12 b	4.65 ± 0.33 c
T6	17.79 ± 0.80 a	1.32 ± 0.11 b	5.22 ± 0.34 b

The values are reported as the mean ± standard deviation of three determinations. Significant differences (*p* < 0.05) are indicated by different letters in the same column. T1–T6 details can be seen in Table 1.

**Table 7 foods-12-03178-t007:** Effect of antioxidant-infused soda–saline washing on textural properties, expressible drip, and whiteness of mackerel (*A. thazard*) surimi gel in comparison with unwashed mince gel.

Treatment	Breaking Force (g)	Deformation (mm)	Gel Strength (g.mm)	Expressible Drip (%)	Whiteness
T1	107.43 ± 3.34 e	3.39 ± 0.39 c	364.27 ± 39.91 d	31.15 ± 0.54 b	40.32 ± 0.12 d
T2	183.17 ± 4.33 c	5.52 ± 0.27 b	1009.66 ± 32.67 b	18.67 ± 0.28 e	42.98 ± 0.47 a
T3	158.07 ± 8.79 d	5.41 ± 0.02 b	854.46 ± 43.71 c	21.13 ± 0.20 d	42.40 ± 0.72 ab
T4	203.77 ± 2.21 b	6.92 ± 0.01 a	1409.35 ± 14.52 a	17.14 ± 0.31 f	42.48 ± 0.40 ab
T5	223.89 ± 5.52 a	6.36 ± 0.27 a	1422.21 ± 38.12 a	28.85 ± 0.97 c	41.74 ± 0.30 bc
T6	218.15 ± 5.70 a	5.04 ± 0.55 b	1100.67 ± 144.34 b	34.74 ± 0.55 a	41.18 ± 0.62 c

Values are given as mean ± standard deviation from triplicate determinations. Different letters in the same column indicate significant differences (*p* < 0.05). T1–T6 details can be seen in Table 1.

**Table 8 foods-12-03178-t008:** Effect of antioxidant-infused soda–saline washing on lipid oxidation and fishy odor score of mackerel (*A. thazard*) surimi gel in comparison with unwashed mince gel.

Treatment	Peroxide Value (meq/kg Lipid)	Thiobarbituric Acid Reactive Substances (mg MDA Equivalent/kg Sample)	Fishy Odor Score *	Rancid Odor Score *
T1	8.20 ± 0.50 a	0.22 ± 0.04 c	3.00 ± 0.67 a	1.00 ± 1.15 a
T2	2.70 ± 0.00 c	0.25 ± 0.03 bc	2.20 ± 1.14 ab	0.60 ± 1.07 a
T3	1.45 ± 0.25 d	0.24 ± 0.02 bc	2.10 ± 1.37 ab	0.40 ± 0.97 a
T4	0.45 ± 0.25 e	0.25 ± 0.06 bc	1.40 ± 0.84 b	0.20 ± 0.63 a
T5	2.45 ± 0.35 c	0.28 ± 0.02 ab	2.20 ± 1.23 ab	0.60 ± 1.26 a
T6	5.20 ± 0.87 b	0.30 ± 0.01 a	1.70 ± 0.82 b	0.60 ± 1.07 a

Values are given as mean ± standard deviation from triplicate determinations except for off-odor score (*n* = 10). Different letters in the same column indicate significant differences (*p* < 0.05). * A score of 4 represented “very strong” while 0 represented “none”. MDA = malondialdehyde. T1–T6 details can be seen in Table 1.

## Data Availability

Data are contained within the article.

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
