# Peer review of "Impact of Washing with Antioxidant-Infused Soda–Saline Solution on Gel Functionality of Mackerel (Auxis thazard) Surimi"

_foods, 2023, doi:10.3390/foods12173178_

Round 1

Reviewer 1 Report

The manuscript submitted to the Foods Journal introduces an innovative approach to washing mackerel surimi using cold carbonated water with NaCl and antioxidants. The study thoroughly covers lipid and myoglobin removal, biochemical characteristics, gelling properties, and oxidative stability, with a detailed analysis of different treatments. Strengths of the paper include its originality, comprehensive experimental design, clear data presentation, quality analysis using scanning electron microscopy, and practical industrial implications, especially with treatment T4. However, some weaknesses need to be addressed. The abstract lacks sufficient contextual information and a clear statement of the problem. The study's introduction should emphasize its innovation, detailing what sets it apart from previous research. In the methodology section of the study, it appears that information regarding the repetitions of analyses is missing. Including details about how many times the analyses were repeated would enhance the clarity and reproducibility of the methods, thus strengthening the overall validity of the research. The study's results are presented clearly and well-discussed, contributing to the overall understanding of the findings. However, an area for improvement lies in the resolution of the figures. Enhancing the image quality would facilitate better visual comprehension and add value to the overall presentation of the research. The conclusion effectively summarizes the research's key findings, emphasizing the improvement in mackerel surimi's gel-forming ability and oxidative stability. It is well-structured and highlights potential practical implications, making it suitable for this study. The manuscript holds merit and offers valuable insights into its field of study; it is suitable for following the necessary corrections and adjustments.

Minor editing of English language required.

Author Response

Reviewer 1

The manuscript submitted to the Foods Journal introduces an innovative approach to washing mackerel surimi using cold carbonated water with NaCl and antioxidants. The study thoroughly covers lipid and myoglobin removal, biochemical characteristics, gelling properties, and oxidative stability, with a detailed analysis of different treatments. Strengths of the paper include its originality, comprehensive experimental design, clear data presentation, quality analysis using scanning electron microscopy, and practical industrial implications, especially with treatment T4. However, some weaknesses need to be addressed.

Ans: Thank you very much.

The abstract lacks sufficient contextual information and a clear statement of the problem.

Ans: It was stated in the beginning of the Abstract that “Mackerel (Auxis thazard), a tropical dark-fleshed fish, has the potential to be used in the production of surimi. It is necessary to identify the optimal washing method to make better use of this species since efficient washing is the most important step in surimi processing to ensure maximal gelling and high-quality surimi.”

The study's introduction should emphasize its innovation, detailing what sets it apart from previous research.

Ans: The Introduction was written to provide research background, state the research question, and propose a possible solution to such challenges. A thorough literature review was undertaken to support up the research statement.

In the methodology section of the study, it appears that information regarding the repetitions of analyses is missing. Including details about how many times the analyses were repeated would enhance the clarity and reproducibility of the methods, thus strengthening the overall validity of the research.

Ans: It was stated in the Statistical Analysis section that “Throughout this study, a completely randomized design was adopted. All experiments were carried out in triplicate (n = 3) except for the off-odor score which was evaluated by 10 trained panelists. The data were reported by mean ± standard deviation. Data were subjected to ANOVA, and mean comparisons were performed using Duncan's Multiple Range Test. The statistical program (SPSS) was used for the statistical analysis.” All of the methodologies employed in this study were both standard and published procedures in international journals, which can be retrieved for additional information.

The study's results are presented clearly and well-discussed, contributing to the overall understanding of the findings. However, an area for improvement lies in the resolution of the figures. Enhancing the image quality would facilitate better visual comprehension and add value to the overall presentation of the research.

Ans: The quality and resolution of the Figures were prepared in accordance with the journal's requirements.

The conclusion effectively summarizes the research's key findings, emphasizing the improvement in mackerel surimi's gel-forming ability and oxidative stability. It is well-structured and highlights potential practical implications, making it suitable for this study. The manuscript holds merit and offers valuable insights into its field of study; it is suitable for following the necessary corrections and adjustments.

Ans: Thank you very much.

Reviewer 2 Report

Abstract:

Line 8:  Need to mention sensory analysis  (fishy odor).

Line 17:  Change the term of "dramatically" with scientific term such as "decrease significantly (p<0.05) or (p<0.01)".

Line 18:  State number of panellist involved.

Introduction:

Additional related reference on effect of washing on surimi-like material:

Gelation properties of spent-duck meat surimi-like material produced using acid-alkaline solubilization methods. Journal of Food Science. 2011, 76 (1): S48-S55.

Line 125-127:  Need to provide details objectives of the study, number of treatment  and analysis conducted.  

Materials and Methods:

Line 135:  Need to mention details of dressed procedures of fish sample.

Line 153:  Need to mention the operation temperature and length of air-blast freezer conducted.

Line 221: Change the term of "estimated" to "calculated"

Line 227-231: Need to provide reference(s) used. Need to mention the details of training session conducted.

Result and Discussion:

Line 246:  Better state the yield for the treatment  T1 (100 %).

Line 249-254:;  Need to mention the pH of conventional or commercial surimi.

Line 255-261:  Need to mention whether the resulted Reactive sulfhydryl content within the normal range in surimi or surimi-like material?

Line 262-274:  Need to compare the resulted Ca2+-ATPase activity with published reported data from surimi or surimi-like material.

Line 275-line 281:  Need to mention whether the resulted Surface hydrophobicity content within the normal range in surimi or surimi-like material?

Line 294:  Need to mention whether reduction of 65-76% consider higher, average or lower than previous reported study? including with sample surimi-like material,

Line 309:  Need to compare the resulted data of PV, CD and TBARS  with the fish surimi of non-fish surimi (surimi lime product).

Line 354-359:  Need to mention whether resulted myoglobin within the normal range or lower or higher?

Line 360-369:  Compare the result of heme and non heme iron of resulted mackerel surimi  with other fish surimi  of surimi line materials.

 Line 370-376:  Need to mention whether resulted metmyoglobin within the normal range or lower or higher?

Line 383-396:  Compare the resulted of rheology data with previous published article of fish surimi of surimi-like materials.

Line 403-424:  Need to mention whether resulted myoglobin within the normal range or lower or higher?

Line 442-456.  Compare the result of textural properties with the textural properties of fish surimi of non fish surimi (surimi-like materials).

Line 492-502:  Need to mention the commercial value of peroxide value sample with fish surimi pr surimi like materials.

Line 521:  Table 8.  Need to correlate the resulted data with the previous commercial fish surimi or non fish surimi,  (Peroxide value and TARS).

Conclusion: Too short conclusion. Need to mention the code of samples (T1 to T6)  and some result of the analysis conducted.

References:  Not provided

Author Response

Reviewer 2

Abstract:

Line 8:  Need to mention sensory analysis  (fishy odor).

Ans: Sensory analysis was mentioned. The purpose of this study was to evaluate the combined effect of cold carbonated water (CW) with NaCl and antioxidants in washing media, so-called antioxidant-infused soda-saline solution, on lipid and myoglobin removal efficacy, biochemical characteristics, gelling properties, sensory features, and the oxidative stability of mackerel (Auxis thazard) surimi in comparison with unwashed mince (T1) and conventional water washed surimi (T2).

Line 17:  Change the term of "dramatically" with scientific term such as "decrease significantly (p<0.05) or (p<0.01)".

Ans: Done. “After washing, lipid decreased significantly (p < 0.05), accounted for a 65-76% reduction.”

Line 18:  State number of panellist involved.

Ans: Done. “Because of the low level of TBARS, all 10 panelists rated rancid odor as low (~1 out of 4), with no significant variations across treatments.

Introduction:

Additional related reference on effect of washing on surimi-like material:

Gelation properties of spent-duck meat surimi-like material produced using acid-alkaline solubilization methods. Journal of Food Science. 2011, 76 (1): S48-S55.

Ans: Related references regarding the effect of washing on surimi-like materials were added.

  1. Nurkhoeriyati, T.; Huda, N.; Ahmad, R. Gelation properties of spent duck meat surimi‐like material produced using acid–alkaline solubilization methods. Food Sci. 2011, 76(1), S48-S55.
  2. Ramadhan, K.; Huda, N.; Ahmad, R. Effect of number and washing solutions on functional properties of surimi-like material from duck meat. Food Sci. Technol. 2014, 51, 256-266.
  3. Chaijan, M.; Srirattanachot, K.; Panpipat, W. Biochemical property and gel‐forming ability of surimi‐like material from goat meat. J. Food Sci. Technol. 2021, 56(2), 988-998.

Line 125-127:  Need to provide details objectives of the study, number of treatment  and analysis conducted.  

Ans: The objective, the number of treatments, and analysis performed were all mentioned. “As a result, the objective of this study was to highlight a novel method based on soda-saline washing combined with antioxidants to improve the removal efficiency of myoglobin and lipid from mackerel (A. thazard) mince and the overall quality of the resulting surimi. Six treatments (T1-T6) were carried out. Unwashed mince (T1) and conventional water washed surimi (T2) were compared with mackerel mince washed with CW in the presence of 0.6% NaCl at a medium to mince ratio of 3:1 (v/w) without antioxidant (T3) or with the addition of 1.5 mM EDTA plus 0.2% (w/v) sodium erythorbate and 0.2% sodium tripolyphosphate (T4), 100 mg/L gallic acid (T5), and 5 mM citric acid containing 8 mM calcium chloride (T6).

Materials and Methods:

Line 135:  Need to mention details of dressed procedures of fish sample.

Ans: The detail was added. “The fish were then headed, eviscerated, washed, filleted, and skinned, respectively. Then, the fish meat was uniformly minced using a meat grinder (a 4-mm hole diameter; Panasonic MK-G20MR, Japan).

Line 153:  Need to mention the operation temperature and length of air-blast freezer conducted.

Ans: The detail was given. “All samples were mixed thoroughly with 4% sucrose and 4% sorbitol before being frozen in an air-blast freezer (Polar DN494 367 Blast Freezer, Campbell town, NSW, Australia) at -21 °C for 1 h. Before being used for analysis, the frozen samples were stored -18 ± 2°C. The frozen storage period was no more than one month.

Line 221: Change the term of "estimated" to "calculated"

Ans: Done.

Line 227-231: Need to provide reference(s) used. Need to mention the details of training session conducted.

Ans: References were provided, and the sensory evaluation was detailed. “Ten trained panelists (5 females and 5 males) with ages between 23 to 35 years from the laboratory staff and graduate students of the Food Technology and Innovation Research Center of Excellence, School of Agricultural Technology and Food Industry, Walailak University, performed sensory examination of gel. The panelists were regular surimi consumers with no surimi allergies. The panelists were chosen based on their sensitivity to fishy and rancid odors and their extensive expertise in assessing the off-odors and off-flavors of cooked seafood and surimi gel. All of them were trained to have rich professional knowledge on fishy and rancid odor [4]. Gel samples were cut to a thickness of 1 cm and a diameter of 2.5 cm, equilibrated at room temperature for 30 min, and coded with 3-digit random numbers. Before the sensory evaluation, samples were kept in a plastic cup with a cover. After opening the cup, the panelists were instructed to sniff the headspace above the samples. The intensity of fishy and rancid odors was graded on a 5-point scale varying between none (score = 0) through strong (score = 4) [54]. The Walailak University Human Research Ethics Committee (WUEC-21-125-02) authorized the study's protocol.

Result and Discussion:

Line 246:  Better state the yield for the treatment  T1 (100 %).

Ans: Done.

Line 249-254:;  Need to mention the pH of conventional or commercial surimi.

Ans: It was stated that “The pH of conventional surimi (T2) was 6.16.”

Line 255-261:  Need to mention whether the resulted Reactive sulfhydryl content within the normal range in surimi or surimi-like material?

Ans: It was stated that “The SH level of frigate mackerel surimi made by ultrasound-aided washing was around 6 mol/108 g protein [4, 55], which was near to the value reported herein, but higher SH content can be found in surimi-like material from spent duck meat [16] and goat meat [18].

Line 262-274:  Need to compare the resulted Ca2+-ATPase activity with published reported data from surimi or surimi-like material.

Ans: It was stated that “The Ca2+-ATPase activity of conventional surimi from the same fish species was reported to be 1.79-2.10 mmolPi/mg protein/min [4, 55], which was within the range of the value found in this study.

Line 275-line 281:  Need to mention whether the resulted Surface hydrophobicity content within the normal range in surimi or surimi-like material?

Ans: It was stated that “The surface hydrophobicity of mackerel surimi produced by single washing was reported as 41.42 mg of BPB bound [4], which was roughly comparable to the values observed in this study.

Line 294:  Need to mention whether reduction of 65-76% consider higher, average or lower than previous reported study? including with sample surimi-like material,

Ans: It was mentioned that “The lipid reduction was within the range of the manufacture of surimi-like material from duck meat using 3-cycle washing with different media (tap water, NaCl, sodium bicarbonate, and sodium phosphate buffer), which eliminated the lipids between 62.5 and 78.6% [17].

Line 309:  Need to compare the resulted data of PV, CD and TBARS  with the fish surimi of non-fish surimi (surimi lime product).

Ans: We mentioned the permissible level of TBARS and it was stated that “According to Larouche et al. [62], lipids are categorized as not oxidized (TBARS value < 1.5 mg MDA/kg), moderately oxidized (1.6 < TBARS value < 3.6), or oxidized (TBARS value > 3.7). Thus, all of the surimi in this study were within the permissible requirements for the TBARS level.

Line 354-359:  Need to mention whether resulted myoglobin within the normal range or lower or higher?

Ans: It was stated that “Depending on the type of raw materials used and the washing techniques employed, it was reported that surimi had lower myoglobin, heme iron, and non-heme iron levels than unwashed mince [4, 16, 17, 18, 51, 55].

Line 360-369:  Compare the result of heme and non heme iron of resulted mackerel surimi  with other fish surimi  of surimi line materials.

Ans: It was stated that “Depending on the type of raw materials used and the washing techniques employed, it was reported that surimi had lower myoglobin, heme iron, and non-heme iron levels than unwashed mince [4, 16, 17, 18, 51, 55].

 Line 370-376:  Need to mention whether resulted metmyoglobin within the normal range or lower or higher?

Ans: It was stated that “In this investigation, the metmyoglobin content of all surimi washed with antioxidant-infused soda-saline solutions was lower than that of frigate mackerel washed conventionally, as reported by Somjid et al. [4].

Line 383-396:  Compare the resulted of rheology data with previous published article of fish surimi of surimi-like materials.

Ans: It was stated that “The rheological properties of all surimi washed with antioxidant-infused soda-saline solutions were similar to those of other dark-fleshed fish surimi, however the values of G', G", and tan δ differed to some extent [4].

Line 403-424:  Need to mention whether resulted myoglobin within the normal range or lower or higher?

Ans: The myoglobin content was discussed previously. “Depending on the type of raw materials used and the washing techniques employed, it was reported that surimi had lower myoglobin, heme iron, and non-heme iron levels than unwashed mince [4, 16, 17, 18, 51, 55].

Line 442-456.  Compare the result of textural properties with the textural properties of fish surimi of non fish surimi (surimi-like materials).

Ans: It was stated that “The textural characteristics of all the surimi washed with antioxidant-infused CW-saline solutions were relatively similar to those of other dark-fleshed fish surimi, but were nevertheless lower than those of other animal surimi-like materials [4, 16, 17, 18].”

Line 492-502:  Need to mention the commercial value of peroxide value sample with fish surimi pr surimi like materials.

Ans: It was stated that “According to Ghimire et al. [71], the PV threshold limit in beef meatballs during refrigerated storage was set at 1 meq peroxide/kg fat. T4 was considered an effective medium in retarding the development of primary lipid oxidation product in mackerel surimi due to that permitted level of PV.

Line 521:  Table 8.  Need to correlate the resulted data with the previous commercial fish surimi or non fish surimi,  (Peroxide value and TARS).

Ans: It was stated that “The TBARS level in all of the surimi gel in the present study was within the permissible range (TBARS < 1.5 mg MDA/kg) [62].

Conclusion: Too short conclusion. Need to mention the code of samples (T1 to T6)  and some result of the analysis conducted.

Ans: The conclusion was concise and informative. The terms "gel forming ability" and "oxidative stability" were used to describe all of the aspects that were investigated. “The incorporation of antioxidant, particularly 1.5 mM EDTA+0.2% sodium erythorbate+0.2% sodium tripolyphosphate) in CW containing 0.6% NaCl, can be the strategy in improving the gel-forming ability and oxidative stability of mackerel (A. thazard) surimi. This medium can be referred to as the lipid-stabilizing technique, which was created to reduce oxidation and rancidity while also improving gel functionality in dark-fleshed fish. The findings from this study will aid in establishing the feasibility of mackerel, a dark-flesh fish species, as a raw material for surimi manufacturing, which will benefit the surimi industry's sustainability. Further study can be conducted to investigate the storage stability of surimi made using this washing medium in order to increase its industrial application.
